# Barron's Theorem for Equivariant Networks

**Hannah Lawrence**                                                    HANLAW@MIT.EDU

*Massachusetts Institute of Technology, Cambridge, MA 02139*

**Editors:** Sophia Sanborn, Christian Shewmake, Simone Azeglio, Arianna Di Bernardo, Nina Miolane

## Abstract

The incorporation of known symmetries in a learning task provides a powerful inductive bias, reducing the sample complexity of learning equivariant functions in both theory and practice. Group-symmetric architectures for equivariant deep learning are now widespread, as are accompanying universality results that verify their representational power. However, these symmetric approximation theorems suffer from the same major drawback as their original non-symmetric counterparts: namely, they may require impractically large networks. In this work, we demonstrate that for some commonly used groups, there exist smooth subclasses of functions — analogous to Barron classes of functions — which can be *efficiently* approximated using invariant architectures. In particular, for permutation subgroups, there exist invariant approximating architectures whose sizes, while dependent on the precise orbit structure of the function, are in many cases just as small as the non-invariant architectures given by Barron's Theorem. For the rotation group, we define an invariant approximating architecture with a new invariant nonlinearity, which may be of independent practical interest, that is similarly just as small as its non-invariant counterparts. Overall, we view this work as a first step towards capturing the *smoothness* of invariant functions in invariant universal approximation, thereby providing approximation results that are not only invariant, but *efficient*.

**Keywords:** Equivariance, invariance, symmetry, universality, approximation, Barron

## 1. Introduction

Symmetries arise in a wide range of learning tasks, with symmetry groups ranging from the discrete (such as permutations of sets or graph nodes) to the continuous (such as rotations of point clouds). In such settings, restricting the hypothesis class to only symmetric functions improves generalization bounds in linear and kernel learning (Elesedy and Zaidi, 2021; Mei et al., 2021), and empirically aids generalization in deep learning (Cohen and Welling, 2016; Kondor et al., 2018). Group-symmetric architectures come in many shapes and sizes, however, and one can ask how expressive various architectures truly are.

Yarotsky (2018) first demonstrated that shallow neural nets, based on classical results on polynomial invariants and equivariants (Hilbert, 1890, 1893), are universal approximators of invariant and equivariant continuous functions, respectively, for compact groups. Later works built on this framework for many different groups: for permutation subgroups, Maron et al. (2019b) defined tensor networks, and proved universality when the network has sufficiently high order tensor activations. Similarly, Maron et al. (2019a) defined invariant graph networks, which are universal for certain classes of graph functions (dependent on the network's tensor order). Bogatskiy et al. (2020) defined universal tensor-product networks equivariant to the Lorentz group, while Dym and Maron (2021) showed that commonly used

point cloud architectures are universal. DeepSets, a permutation-invariant architecture for learning on sets, has also been shown to be universal (Zaheer et al., 2017).

However, these results do not bound the *size* of the network needed to approximate a continuous equivariant function arbitrarily well. Indeed, the size of the neural network will often depend at least exponentially on the input dimension, or on abstract algebraic quantities such as the cardinality or degree of the generating set of polynomials. In fact, this caveat dates back to the original universality theorems by Cybenko (Cybenko, 1989) and Hornik (Hornik et al., 1990), who first demonstrated that shallow neural networks with sigmoidal activations are universal approximators. These early universality results relied on polynomial approximation, yielding exponential dependence on dimension. In his seminal work, Barron (1993) circumvented this drawback by defining a class of smooth functions which can be approximated *efficiently*, in terms of network size. Yet, an analogous result for invariant networks has not yet been shown. Our central question is therefore: *What classes of invariant functions can be* efficiently *approximated by invariant neural networks?*

Of course, one can trivially group-average any approximating architecture, such as that of Barron (1993), to obtain an invariant approximation with error at least as good (Yarotsky, 2018). However, the size of the network naively increases by a factor of $|G|$, which may be large or even infinite. (Here, we define the size of the network by the number of hidden units, rather than the number of learnable parameters. Indeed, group-averaging does not increase the number of learnable parameters, but it significantly increases the number of hidden units in the network, and therefore the computational complexity of a forward pass.) Our main contribution is to demonstrate that this blow-up in size is not necessary for several commonly used groups and function classes.

**Our Contributions** For subgroups of the permutation group $S_n$ (Theorem 2), we show that there exist *invariant* approximating architectures whose size very often improves on the group-averaging baseline, and in many non-trivial cases by a factor of $|G|$. For the rotation group (Theorem 7), we analytically simplify the group-averaging operation to obtain an invariant approximating architecture with a novel analytic nonlinearity. As a result, group-averaging need not increase the number of nodes in the network.

**Preliminaries and Notation** Let $G$ be a group acting orthogonally on $\mathcal{X} \subseteq \mathbb{R}^n$. A function $f : \mathcal{X} \to \mathbb{R}$ is $G$-invariant if $f(\boldsymbol{x}) = f(g\boldsymbol{x}) \, \forall \boldsymbol{x} \in \mathcal{X}, g \in G$. $\mathcal{X}$ is partitioned into possibly unequally-sized[1] orbits $\mathcal{O}$, where $[\boldsymbol{x}] = \{g\boldsymbol{x} : g \in G\} \in \mathcal{O}$. $\boldsymbol{x}_i$ is the $i^{th}$ entry of $\boldsymbol{x}$.

**Barron's Theorem** We review Barron's Theorem below; see Appendix A.1 for details.

**Theorem 1 ( Barron (1993))** *Let $f : \mathbb{R}^n \to \mathbb{R}$ have Fourier transform $\hat{f}$ and "Barron parameter" $C_f := \int_{\boldsymbol{w} \in \mathbb{R}^n} \|\boldsymbol{w}\| \cdot |\hat{f}(\boldsymbol{w})|$. Then, there exists a two-layer neural network with nonlinearity $\sigma(x) = \mathbb{1}(x \geq 0)$, and approximation error $||f - NN_{Barron}||_2^2$ at most $\frac{\Theta(C_f^2)}{k}$.*

The central idea of Barron's Theorem is to sample the Fourier basis functions[2] proportionally to $f$'s Fourier coefficients, which produces a two-layer cosine[3] network. Maurey's

---

1. For example, the orbit of $[1, 1, \ldots, 1]$ over $S_n$ is itself, while the orbit of $[1, 2, \ldots, n]$ has size $|S_n| = n!$.
2. In Section 2, we make the minor switch to discrete (rather than continuous) Fourier series, which simplifies the analysis for invariant functions while retaining the key intuitions of Barron's Theorem.
3. An algebraic manipulation transforms the cosine nonlinearity to an indicator.

Lemma (Pisier, 1981) (see Lemma 15) then bounds the network size versus approximation error tradeoff, with error governed by the expected norm of the sampled functions.

## 2. Subgroups of $S_n$

Let $G \leq S_n$, and throughout this subsection, let $f : [0,1]^n \to \mathbb{R}$ be a $G$-invariant function. $f$ can be decomposed using discrete Fourier series:

$$f(\boldsymbol{x}) = \sum_{\boldsymbol{w} \in \mathbb{Z}^n} \hat{f}(\boldsymbol{w})e^{2\pi i\langle \boldsymbol{x}, \boldsymbol{w}\rangle} = \sum_{[\boldsymbol{w}] \in \mathcal{O}} \sum_{\boldsymbol{w} \in [\boldsymbol{w}]} \mathrm{Re}(\hat{f}(\boldsymbol{w}))\cos(2\pi\langle \boldsymbol{x}, \boldsymbol{w}\rangle) - \mathrm{Im}(\hat{f}(\boldsymbol{w}))\sin(2\pi\langle \boldsymbol{x}, \boldsymbol{w}\rangle),$$

where $\hat{f}(\boldsymbol{w}) = \int_{\boldsymbol{x} \in [0,1]^n} f(\boldsymbol{x})e^{-2\pi i\langle \boldsymbol{x}, \boldsymbol{w}\rangle}$. We define an inner product over periodic functions, $\langle f, g\rangle = \int_{[0,1]^n} f(\boldsymbol{x})g(\boldsymbol{x})$. It is easy to check that $\hat{f}(\boldsymbol{w}) = \hat{f}(g\boldsymbol{w}) \ \forall g \in G, \forall \boldsymbol{w} \in \mathbb{Z}^n$ (Appendix A.2). Using this Fourier invariance structure, we will see that Theorem 1 can be non-trivially extended for invariant networks by sampling *orbits*, instead of individual frequencies, in Fourier space. In fact, this can be done for any invariant function and group; naively, it is equivalent to group-averaging the architecture of Barron (1993). For permutation subgroups, however, one can show that fewer sampled orbits than group-averaging are needed to achieve the original network's approximation error, thereby circumventing the blow-up in size.

**Theorem 2** *Let $\mathcal{N} = \sum_{\boldsymbol{w} \in \mathbb{Z}^n} |\hat{f}(\boldsymbol{w})|$, and let $m$ and $M$ denote the minimum and maximum, respectively, orbit sizes $|[\boldsymbol{w}]|$ in $\mathrm{Supp}(\hat{f})$. Then there exists a two-layer group-invariant network $NN$ with cosine activations and $kM$ hidden units such that*

$$\int_{[0,1]^n} |f(\boldsymbol{x}) - NN(\boldsymbol{x})|^2 d\boldsymbol{x} \leq \frac{\Theta(\mathcal{N} \cdot \sum_{[\boldsymbol{w}] \in \mathcal{O}} |\hat{f}([\boldsymbol{w}])|)}{k} \leq \frac{\Theta(\mathcal{N}^2)}{mk} \tag{1}$$

The proof of Theorem 2 (see Appendix A.3) modifies that of Theorem 1 by sampling orbits instead of individual frequencies in Fourier space, to ensure invariance. It proceeds by approximating the even and odd parts of $f$ separately; then, the discrete cosines and sines, are orthogonal within a given orbit, and the squared norm of each orbit (which controls the approximation error from sampling) can be decomposed with the Pythagorean theorem.

**Remark 3 (Comparison to Barron)** *A non-invariant version of Theorem 2 would obtain error $\Theta\left(\frac{\mathcal{N}^2}{k}\right)$ with $k$ hidden units; then, naive group-averaging would obtain error $\Theta\left(\frac{\mathcal{N}^2}{k}\right)$ with $kM$ units. Therefore, up to a constant factor, Theorem 2 has a factor of $m$ improvement over group-averaging. Moreover, if $m = M$ (all orbits are equally sized), there is no increase in size relative to the non-invariant approximation (in the regime where the original approximation error is sufficiently small, i.e. at most $\Theta\left(\frac{\mathcal{N}^2}{M}\right)$).*

**Remark 4 (Comparison to tensor networks)** *Although not directly comparable, this approximation result has several distinctive features from that of Maron et al. (2019b). First, the tensor networks of Maron et al. (2019b) rely on both approximating invariant polynomials, and universal approximation of the multiplication function, as key primitives*

to achieving universality. These would induce an exponential dependence on $n$ in an end-to-end approximation guarantee, avoided by Theorem 2. Second, the intermediate tensor order from Maron et al. (2019b) may be as high as $n^{\frac{n(n-1)}{2}}$ (dependent on $G$ but not approximation error), whereas for approximation error $\Theta(\frac{\mathcal{N}^2}{m})$ (therefore constant $k$), the group-invariant networks of Theorem 2 have size scaling with orbit size[4].

Although already an improvement for many functions and groups, Theorem 2 does not improve over group-averaging when $m = 1$, and is more generally loose when $m$ is much smaller than $M$. A more precise version of Theorem 2 below (proven in Appendix A.4) articulates the exact interplay between $f$'s orbit structure and approximation error. For simplicity of presentation, we assume that $f$ is even ($f(\boldsymbol{x}) = f(1 - \boldsymbol{x})$), but an analogous statement holds for odd functions as well (and therefore for arbitrary invariant functions).

**Theorem 5** *Let $f$ be an even function, $v_{[\boldsymbol{w}]} = \sum_{\boldsymbol{w} \in [\boldsymbol{w}]} sgn(\hat{f}(\boldsymbol{w})) \cos(2\pi \langle \boldsymbol{w}, \boldsymbol{w} \rangle)$, $x_1$ equal $\mathcal{N} \cdot v_{[\boldsymbol{w}]}$ with probability $\frac{|\hat{f}([\boldsymbol{w}])| \cdot |[\boldsymbol{w}]|}{\mathcal{N}}$, and $s_1$ equal the corresponding orbit size $|[\boldsymbol{w}]|$. Then there exists a two-layer group-invariant network $NN$ with cosine activations, equal to a weighted sum of $k$ orbits $v_{[\boldsymbol{w}]}$ and comprising $H$ total hidden units, such that*

$$\int_{[0,1]^n} |f(x) - NN(x)|^2 dx = \Theta\Big((2 - k)\mathbb{E}[s_1]||f||^2 - 2\langle f, \mathbb{E}[x_1 s_1] \rangle + \mathcal{N}^2 \mathbb{E}[s_1^2] + (k - 1)\mathbb{E}[s_1]^2 \mathcal{N}^2\Big)$$

**Remark 6** *Note that Theorem 5 depends on the new quantities $\mathbb{E}[s_1]$, $\mathbb{E}[s_1^2]$, and $\mathbb{E}[x_1 s_1]$, which capture how the mass of $\hat{f}$ is distributed among orbits of different sizes. If most or all of the mass of $\hat{f}$ is on equal-sized orbits, Theorem 5 essentially recovers Theorem 2 with $m = M$.*

## 3. Rotations

As noted in Section 1, one straightforward approach to obtaining universal approximation theorems for invariant/equivariant architectures is simply to group-average a generic approximation, such as that of Theorem 1, producing a symmetric architecture with no worse approximation error (Appendix A.5). However, this architecture is naively a factor of $|G|$ larger than the original architecture. In the previous section, we used a tighter sampling analysis to avoid this blow-up for permutation subgroups. In this section, we instead look closely at the group-averaged network architecture for $G = SO(n)$, and show that it can be computed with a network whose size is *no larger* than that of the original architecture, by careful choice of nonlinearity.

**Theorem 7** *When averaged over the special orthogonal group in $n$ dimensions, the nonlinear invariant function $\int_{g \in SO(n)} \mathbb{1}(\boldsymbol{w}^T g \boldsymbol{x} - b \geq 0)$ is expressible as follows:*

$$\bar{\mathbb{1}}_{\boldsymbol{w}, b}(x) := \int_{g \in SO(n)} \mathbb{1}(\boldsymbol{w}^T g \boldsymbol{x} - b \geq 0) = \frac{1}{2} I_{\sin(\phi)^2}\Big(\frac{n - 1}{2}, \frac{1}{2}\Big).$$

---

4. Although the orbit size may be as large as $|G|$, this quantity is dwarfed by $n^{\frac{n(n-1)}{2}}$ asymptotically.

Here, $\phi = \cos^{-1}(\frac{b}{||\boldsymbol{w}|| \cdot ||\boldsymbol{x}||})$ and $I_a(b, c)$ denotes the regularized incomplete beta function. As a result, any network (including that of Theorem 1) of the form $\boldsymbol{\alpha}^T \mathbb{1}(\mathbf{W}\boldsymbol{x} - \boldsymbol{b})$ can be averaged over $SO(n)$ using regularized incomplete beta function nonlinearities, with no further increase in network size.

The proof of Theorem 7 is in Appendix A.6. To our knowledge, no $SO(n)$-invariant architecture have used such a nonlinearity before. Computing analytic forms for invariant and equivariant network layers is itself an exciting direction for future work, both for stronger approximation results and as a practical tool in equivariant architecture design.

**Reducing to orbit space** Instead of averaging a non-invariant architecture over $SO(n)$ as in Theorem 7, one could also compute the smooth orbit mapping $x \mapsto \|x\|_2$, and then apply any universal approximation theorem (such as Barron's Theorem) in one dimension to approximate an $SO(n)$-invariant function. (Indeed, this approach is feasible for other groups using any method for mapping to orbits, such as Dym and Gortler (2022).) The resultant neural network may be more efficient (if the Barron parameter in the radial dimension is smaller than in the ambient space), although this does not provide a method for precisely group-averaging any existing one-layer architecture.

## 4. Conclusion

In this work, we provided evidence that invariant, universal, and small architectures should exist for broad classes of "nice" invariant functions. Barron's Theorem (Barron, 1993) first provided a particular quantification of the smoothness of a function, based on how well it can be approximated by sampling Fourier coefficients; here, we adapt this notion of smoothness for group-invariant functions, instead structuring our sampling in Fourier space to produce an approximating architecture that is itself invariant, too. However, this notion of smoothness arose as a consequence of the particular technical tools of Barron's Theorem. Is there a better, more inherent way of measuring the smoothness of invariant functions? For example, as noted in the previous section, the orbits of a radially symmetric function correspond to $\mathbb{R}_+$, and the function's approximability via neural nets can also be governed by its smoothness along the radial dimension: rigorously extending this intuition to other groups is an exciting direction for future work. Overall, we hope that the techniques and examples developed here serve as an invitation to further develop the theory of efficient invariant approximation.

## Acknowledgments

We thank Ankur Moitra for inspiring this work's central question and for advice throughout, and Bobak Kiani for helpful discussions early in the development of the project. HL is supported by the Fannie and John Hertz Foundation and the National Science Foundation Graduate Research Fellowship under Grant No. 1745302.

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

## Appendix A. Proofs

### A.1. Barron's Theorem (original)

**Theorem 8 (Barron's Theorem, Barron (1993))** *Let $f : \mathbb{R}^n \to \mathbb{R}$ be a continuous function with Fourier transform $\hat{f}$ and "Barron parameter" $C_f := \left( \int_{w \in \mathbb{R}^n} \|w\| \cdot |\hat{f}(w)| \right)^2$. Let $\sigma(x)$ be a sigmoidal[5] nonlinearity. In particular, we will consider $\sigma(x) = \mathbb{1}(x \geq 0)$. Then, there exists a two-layer neural network $NN_{Barron}(x) = \sum_{i=1}^{k} \alpha_i \sigma(\langle w_i, x \rangle - b_i)$ with bounded approximation error:*

$$\int |f(x) - NN_{Barron}(x)|^2 \leq \frac{\Theta(C_f)}{k}$$

**Proof** Write $f$ in terms of its Fourier coefficients as follows:

$$f(x) = \text{Re}\left( \int_w e^{2\pi \langle w, x \rangle} \hat{f}(w) dw \right)$$

$$= \text{Re}\left( \int_w e^{2\pi \langle w, x \rangle + 2\pi \measuredangle(w)} |\hat{f}(w)| dw \right)$$

$$= \int_w \cos(2\pi \langle w, x \rangle + 2\pi \measuredangle(w)) |\hat{f}(w)|$$

---

5. A nonlinearity $\sigma$ is sigmoidal if $\lim_{x \to -\infty} \sigma(x) = 0$ and $\lim_{x \to \infty} \sigma(x) = 1$.

Moreover, we can write $f(x) - f(0)$ as $\int_w (\cos(2\pi\langle w, x\rangle + 2\pi\measuredangle(w)) - \cos(2\pi\measuredangle(w)))|\hat{f}(w)|$, and algebraic manipulation yields

$$\cos(2\pi\langle w, x\rangle + 2\pi\measuredangle(w)) - \cos(2\pi\measuredangle(w)) \propto \int_0^{||w||} \mathbb{1}(w^T x - b \geq 0)\sin(2\pi b + 2\pi\measuredangle(w)).$$

Finally, we apply Maurey's Lemma to argue that subsampling in $w$ and $b$ will not induce too much error:

**Lemma 9 (Maurey's Lemma, Pisier (1981))** *Let $x$ lie in a convex set $\mathcal{V}$. Then there exist $k$ elements $v_1, \ldots, v_k \in \mathcal{V}$ and weights $\alpha_1, \ldots, \alpha_k$ such that*

$$||x - \sum_{i=1}^k \alpha_i v_i|| \leq \frac{\sup_{v\in\mathcal{V}} ||v||_2^2}{k}$$

Applying Maurey's Lemma to the set $\mathcal{V}$ indexed by $w$ and $0 \leq b \leq ||w||$ of functions $\mathbb{1}(w^T x \geq b)$ yields the result. ∎

### A.2. Fourier structure of invariant and equivariant functions

In this subsection, we establish the Fourier structure of both invariant and equivariant functions. We begin with periodic functions, and consider the structure of their discrete Fourier series.

**Lemma 10** *Let $G \leq S_n$, and let $f : [0,1]^n \to \mathbb{R}$ (or alternatively, let $f : \mathbb{R}^n \to \mathbb{R}$ be periodic on $[0,1]^n$. Then $\hat{f}(\boldsymbol{w}) = \hat{f}(g\boldsymbol{w})$ for all $g \in G$, $\boldsymbol{w} \in \mathbb{Z}^n$.*

**Proof** As before, the symmetry of $f$ with respect to $G$ is reflected in its Fourier coefficients as well:

$$\hat{f}(gw) = \int_{x\in[0,1]^n} f(x)e^{2\pi i\langle x, gw\rangle}$$

$$= \int_{x\in[0,1]^n} f(x)e^{2\pi i\langle g^T x, w\rangle}$$

$$= \int_{z\in[0,1]^n} f(gz)e^{2\pi i\langle z, w\rangle}$$

$$= \int_{z\in[0,1]^n} f(z)e^{2\pi i\langle z, w\rangle} = \hat{f}(w)$$

Note that the domain of integration $[0,1]^n$ is invariant under permutations $g$, which allowed the change of variable $z = g^T x$. ∎

Although we only use the discrete Fourier transform in the main body, it is worth noting for future work that this invariance structure in Fourier space holds for non-periodic functions as well.

**Lemma 11 (Fourier characterization of invariant functions)** $f : \mathbb{R}^n \to \mathbb{R}$ *is invariant to a group $G \leq O(n)$ iff the Fourier transform of $f$ is also invariant, i.e. $\hat{f}(\boldsymbol{w}) = \hat{f}(g\boldsymbol{w})$ for all $\boldsymbol{w} \in \mathbb{R}^n$ and for all $g \in G$.*

**Proof**

$$\mathcal{F}(f)(gw) = \int_{x \in \mathbb{R}^n} e^{-2\pi i \langle gw, x \rangle} f(x)$$

$$= \int_{x \in \mathbb{R}^n} e^{-2\pi i \langle w, g^T x \rangle} f(x)$$

$$= \int_{z \in \mathbb{R}^n} e^{-2\pi i \langle w, z \rangle} f(z)$$

$$= \mathcal{F}(f)(w)$$

■

Finally, vector-valued equivariant functions are also equivariant in Fourier space, in the following sense.

**Lemma 12 (Fourier characterization of equivariant functions)** $f : \mathbb{R}^n \to \mathbb{R}^m$ *is equivariant to a group $G \leq O(n)$ iff the entrywise Fourier transform $\hat{f} : \mathbb{R}^n \to \mathbb{R}^m$ is also equivariant, i.e. $g\hat{f}(\boldsymbol{w}) = \hat{f}(g\boldsymbol{w})$ for all $\boldsymbol{w} \in \mathbb{R}^n$ and for all $g \in G$.*

**Proof**

$$\mathcal{F}(f)(gw)_s = \int_{x \in \mathbb{R}^n} e^{-2\pi i \langle gw, x \rangle} f_s(x)$$

$$= \int_{x \in \mathbb{R}^n} e^{-2\pi i \langle w, g^T x \rangle} f_s(x)$$

$$= \int_{z \in \mathbb{R}^n} e^{-2\pi i \langle w, z \rangle} f_s(gz)$$

By the equivariance of $f$, the vector $f(gz)$ is equal to $gf(z)$. Therefore, we can interchange the entrywise integral and the group action to complete the proof:

$$\mathcal{F}(f)(gw)_s = \int_{z \in \mathbb{R}^n} e^{-2\pi i \langle w, z \rangle} f(gz)$$

$$= \int_{z \in \mathbb{R}^n} e^{-2\pi i \langle w, z \rangle} gf(z)$$

$$= g \int_{z \in \mathbb{R}^n} e^{-2\pi i \langle w, z \rangle} f(z)$$

$$= g\mathcal{F}(f)(w)$$

■

**A.3. Proof of Theorem 2**

Consider the proof technique of Barron's Theorem applied in this setting, modified to preserve invariance. We will often work with the orbits of $\mathbb{Z}^n$ with respect to $G$, denoted by $\mathcal{O}$. We denote individual orbits by $[\boldsymbol{w}]$, where $o$ is the orbit representative, i.e. $[\boldsymbol{w}] = \{g\boldsymbol{w} : g \in G\}$. Recall the discrete Fourier decomposition of $f$:

$$f(\boldsymbol{w}) = \sum_{[\boldsymbol{w}]\in\mathcal{O}} \sum_{\boldsymbol{w}\in[\boldsymbol{w}]} \operatorname{Re}(\hat{f}(\boldsymbol{w})) \cos(2\pi\langle\boldsymbol{x},\boldsymbol{w}\rangle) - \operatorname{Im}(\hat{f}(\boldsymbol{w})) \sin(2\pi\langle\boldsymbol{x},\boldsymbol{w}\rangle).$$

We will approximate the terms $\sum_{[\boldsymbol{w}]\in\mathcal{O}} \sum_{\boldsymbol{w}\in[\boldsymbol{w}]} \operatorname{Re}(\hat{f}(\boldsymbol{w})) \cos(2\pi\langle\boldsymbol{x},\boldsymbol{w}\rangle)$ and $\sum_{[\boldsymbol{w}]\in\mathcal{O}} \sum_{\boldsymbol{w}\in[\boldsymbol{w}]} \operatorname{Im}(\hat{f}(\boldsymbol{w})) \sin(2\pi\langle\boldsymbol{x},\boldsymbol{w}\rangle)$ separately. Since we would like to sample functions according to a positive probability measure, we have

$$\operatorname{Re}(\hat{f}(\boldsymbol{w})) \cos(2\pi\langle\boldsymbol{x},\boldsymbol{w}\rangle) = |\operatorname{Re}(\hat{f}(\boldsymbol{w}))| sgn(\operatorname{Re}(\hat{f}(\boldsymbol{w}))) \cos(2\pi\langle\boldsymbol{x},\boldsymbol{w}\rangle)$$

and likewise for the second term. We next show that the trigonometric functions above are orthogonal.

**Lemma 13 (Cosine functions are orthogonal)** *Let $w, z \in \mathbb{R}^n$ with $w_1, z_1 \in \mathbb{Z}$ and $w_1^2 \neq z_1^2$. Let $x_1$ indicate the first entry of a vector $x$, and $x_{2:}$ the rest of the entries. Then the following orthogonality relation holds for arbitrary functions $r, s : \mathbb{R}^{2n-2} \to \mathbb{R}$:*

$$\int_{x\in[0,1]^n} \cos(2\pi(x_1 w_1 + r(x_{2:}, w_{2:}))) \cos(2\pi(x_1 z_1 + s(x_{2:}, z_{2:}))) = 0$$

**Proof** We will heavily use the trigonometric identities $\cos(\alpha + \beta) = \cos(\alpha)\cos(\beta) - \sin(\alpha)\sin(\beta)$.

We will also rely on the standard trigonometric orthogonality statements, assuming $n$ and $m$ are integers with distinct absolute values:

$$\int_{x\in[0,1]} \sin(2\pi nx) \cos(2\pi mx) = 0$$

$$\int_{x\in[0,1]} \cos(2\pi nx) \cos(2\pi mx) = 0$$

$$\int_{x\in[0,1]} \sin(2\pi nx) \sin(2\pi mx) = 0$$

Recall the trigonometric identity $\cos\theta\cos\phi = \frac{\cos(\theta-\phi)+\cos(\theta+\phi)}{2}$. If $|n| \neq |m|$, then $\theta - \phi \neq 0$ and $\theta+\phi \neq 0$, and $\int_{x\in[0,1]} \cos(2\pi(n-m)x) = \int_{x\in[0,1]} \cos(2\pi(n+m)x) = 0$. Similar reasoning using the identities $\sin\theta\sin\phi = \frac{\cos(\theta-\phi)-\cos(\theta+\phi)}{2}$ and $\sin\theta\cos\phi = \frac{\sin(\theta-\phi)+\sin(\theta+\phi)}{2}$ yield the other two equalities.

To apply these one-dimensional identities in our setting, again apply the trigonometric identity for the cosine of a sum to decompose $\cos(2\pi(x_1 w_1 + r(x_{2:}, w_{2:})))$, and likewise for

the analogous term with $z$. Therefore, we have the following:

$$\cos(2\pi(x_1 w_1 + r(x_{2:}, w_{2:}))) \cos(2\pi(x_1 z_1 + s(x_{2:}, z_{2:})))$$

$$= \cos(2\pi x_1 w_1 + 2\pi r(x_{2:}, w_{2:})) \cos(2\pi x_1 z_1 + 2\pi s(x_{2:}, z_{2:}))$$

$$= (\cos(2\pi x_1 w_1) \cos(2\pi r(x_{2:}, w_{2:})) - \sin(2\pi x_1 w_1) \sin(2\pi r(x_{2:}, w_{2:})))\cdot$$

$$(\cos(2\pi x_1 z_1) \cos(2\pi s(x_{2:}, z_{2:})) - \sin(2\pi x_1 z_1) \sin(2\pi s(x_{2:}, z_{2:})))$$

$$= \cos(2\pi x_1 w_1) \cos(2\pi r(x_{2:}, w_{2:})) \cos(2\pi x_1 z_1) \cos(2\pi s(x_{2:}, z_{2:})) -$$

$$\cos(2\pi x_1 w_1) \cos(2\pi r(x_{2:}, w_{2:})) \sin(2\pi x_1 z_1) \sin(2\pi s(x_{2:}, z_{2:})) -$$

$$\sin(2\pi x_1 w_1) \sin(2\pi r(x_{2:}, w_{2:})) \cos(2\pi x_1 z_1) \cos(2\pi s(x_{2:}, z_{2:})) +$$

$$\sin(2\pi x_1 w_1) \sin(2\pi r(x_{2:}, w_{2:})) \sin(2\pi x_1 z_1) \sin(2\pi s(x_{2:}, z_{2:}))$$

Recall that the integral of a separable function is equal to the product of the integrals. Therefore, evaluating each term separately:

$$\int_{x\in[0,1]^n} \cos(2\pi x_1 w_1) \cos(2\pi r(x_{2:}, w_{2:})) \cos(2\pi x_1 z_1) \cos(2\pi s(x_{2:}, z_{2:}))$$

$$= \Big( \underbrace{\int_{x_1\in[0,1]} \cos(2\pi x_1 w_1) \cos(2\pi x_1 z_1)}_{=0} \Big) \Big( \int_{x_{2:}\in[0,1]^{n-1}} \cos(2\pi r(x_{2:}, w_{2:})) \cos(2\pi s(x_{2:}, z_{2:})) \Big)$$

$$\int_{x\in[0,1]^n} \sin(2\pi x_1 w_1) \sin(2\pi r(x_{2:}, w_{2:})) \cos(2\pi x_1 z_1) \cos(2\pi s(x_{2:}, z_{2:}))$$

$$= \Big( \underbrace{\int_{x_1\in[0,1]} \sin(2\pi x_1 w_1) \cos(2\pi x_1 z_1)}_{=0} \Big) \Big( \int_{x_{2:}\in[0,1]^{n-1}} \sin(2\pi r(x_{2:}, w_{2:})) \cos(2\pi s(x_{2:}, z_{2:})) \Big)$$

$$\int_{x\in[0,1]^n} \cos(2\pi x_1 w_1) \cos(2\pi r(x_{2:}, w_{2:})) \sin(2\pi x_1 z_1) \sin(2\pi s(x_{2:}, z_{2:}))$$

$$= \Big( \underbrace{\int_{x_1\in[0,1]} \cos(2\pi x_1 w_1) \sin(2\pi x_1 z_1)}_{=0} \Big) \Big( \int_{x_{2:}\in[0,1]^{n-1}} \cos(2\pi r(x_{2:}, w_{2:})) \sin(2\pi s(x_{2:}, z_{2:})) \Big)$$

$$= \Big( \underbrace{\int_{x_1\in[0,1]} \sin(2\pi x_1 w_1) \sin(2\pi x_1 z_1)}_{=0} \Big) \Big( \int_{x_{2:}\in[0,1]^{n-1}} \sin(2\pi r(x_{2:}, w_{2:})) \sin(2\pi s(x_{2:}, z_{2:})) \Big).$$

■

We can build on this lemma to reason about the orthogonality of cosine basis functions, as shown in the following lemma.

**Lemma 14 (Cosine basis functions are orthogonal)**  *Let $w, z \in \mathbb{Z}^n$ with $w \neq z$ and $w \neq -z$. Then the following orthogonality relation holds:*

$$\int_{x\in[0,1]^n} \cos(2\pi\langle x, w\rangle)) \cos(2\pi\langle x, z\rangle) = 0$$

**Proof** First, suppose there exists some index $i \in [1, \ldots, n]$ such that $w_i^2 \neq z_i^2$. Without loss of generality, let $i = 1$. Then by the previous lemma, Lemma 13, the statement holds with $r(x_{2:}, w_{2:}) = \langle x_{2:}, w_{2:} \rangle$ and likewise for $s$.

Therefore, suppose $w_j^2 = z_j^2$ for all $j$. Since $w \neq z$, there is some $j$ such that $w_j = -z_j$. Without loss of generality, let $j = 1$. Again, let $r(x_{2:}, w_{2:}) = \langle x_{2:}, w_{2:} \rangle$ and likewise for $s$.

We have that

$$
\begin{aligned}
\int_{x_1 \in [0,1]} \cos(2\pi x_1 w_1) \cos(2\pi x_1 z_1) &= \int_{x_1 \in [0,1]} \cos(2\pi x_1 w_1) \cos(2\pi - x_1 w_1) \\
&= \int_{x_1 \in [0,1]} \cos^2(2\pi x_1 w_1) \\
&= \int_{x_1 \in [0,1]} \frac{\cos(4\pi x_1 w_1) + 1}{2} \\
&= \frac{1}{2}.
\end{aligned}
$$

Above, we used the double-angle formula $\cos^2(x) = \frac{\cos(2x)+1}{2}$. Moreover,

$$
\begin{aligned}
\int_{x_1 \in [0,1]} \sin(2\pi x_1 w_1) \sin(-2\pi x_1 w_1) &= \int_{x_1 \in [0,1]} -\sin^2(2\pi x_1 w_1) \\
&= \int_{x_1 \in [0,1]} \cos^2(2\pi x_1 w_1) - 1 \\
&= -\frac{1}{2}.
\end{aligned}
$$

We also have that

$$
\int_{x_1 \in [0,1]} \sin(2\pi m x_1) \sin(2\pi n x_1) = 0
$$

for any integers $m$ and $n$, as $\sin\theta \cos\phi = \frac{\sin(\theta-\phi)+\sin(\theta+\phi)}{2}$.

Now, recall that the proof of Lemma 13 decomposed the integral of interest into a (signed) sum of four terms:

$$
\int_{x \in [0,1]^n} \cos(2\pi x_1 w_1) \cos(2\pi r(x_{2:}, w_{2:}) \cos(2\pi x_1 z_1) \cos(2\pi s(x_{2:}, z_{2:})
$$
$$
= \Big( \underbrace{\int_{x_1 \in [0,1]} \cos(2\pi x_1 w_1) \cos(2\pi x_1 z_1)}_{= \frac{1}{2}} \Big) \Big( \underbrace{\int_{x_{2:} \in [0,1]^{n-1}} \cos(2\pi r(x_{2:}, w_{2:}) \cos(2\pi s(x_{2:}, z_{2:})}_{(*)} \Big)
$$

$$
\int_{x \in [0,1]^n} \sin(2\pi x_1 w_1) \sin(2\pi r(x_{2:}, w_{2:}) \cos(2\pi x_1 z_1) \cos(2\pi s(x_{2:}, z_{2:})
$$
$$
= \Big( \underbrace{\int_{x_1 \in [0,1]} \sin(2\pi x_1 w_1) \cos(2\pi x_1 z_1)}_{= 0} \Big) \Big( \int_{x_{2:} \in [0,1]^{n-1}} \sin(2\pi r(x_{2:}, w_{2:}) \cos(2\pi s(x_{2:}, z_{2:}) \Big)
$$

$$
\int_{x \in [0,1]^n} \cos(2\pi x_1 w_1) \cos(2\pi r(x_{2:}, w_{2:}) \sin(2\pi x_1 z_1) \sin(2\pi s(x_{2:}, z_{2:})
$$
$$
= \Big( \underbrace{\int_{x_1 \in [0,1]} \cos(2\pi x_1 w_1) \sin(2\pi x_1 z_1)}_{= 0} \Big) \Big( \int_{x_{2:} \in [0,1]^{n-1}} \cos(2\pi r(x_{2:}, w_{2:}) \sin(2\pi s(x_{2:}, z_{2:}) \Big)
$$
$$
= \Big( \underbrace{\int_{x_1 \in [0,1]} \sin(2\pi x_1 w_1) \sin(2\pi x_1 z_1)}_{= -\frac{1}{2}} \Big) \Big( \underbrace{\int_{x_{2:} \in [0,1]^{n-1}} \sin(2\pi r(x_{2:}, w_{2:}) \sin(2\pi s(x_{2:}, z_{2:})}_{(**)} \Big).
$$

It remains to show that the term $(*)$ is equal to the term $(**)$. We proceed by induction on the orthogonality of not only the cosine functions, but also the sine functions. First, suppose $w_i = z_i$ for all $i \geq 2$. Then $r(x_{2:}, w_{2:}) = s(x_{2:}, z_{2:}) = \langle x_{2:}, w_{2:} \rangle$. Then:

$$
(*) = \int_{x_{2:} \in [0,1]^{n-1}} \cos(2\pi r(x_{2:}, w_{2:})) \cos(2\pi s(x_{2:}, z_{2:}))
$$
$$
= \int_{x_{2:} \in [0,1]^{n-1}} \cos^2(2\pi \langle x_{2:}, w_{2:} \rangle)
$$
$$
= \int_{x_{2:} \in [0,1]^{n-1}} \frac{\cos(4\pi \langle x_{2:}, w_{2:} \rangle) + 1}{2}
$$
$$
= \frac{1}{2},
$$

and

$$(**) = \int_{x_{2:} \in [0,1]^{n-1}} \sin(2\pi \langle x_{2:}, w_{2:} \rangle) \sin(2\pi \langle x_{2:}, w_{2:} \rangle)$$

$$= \int_{x_{2:} \in [0,1]^{n-1}} \sin^2(2\pi \langle x_{2:}, w_{2:} \rangle)$$

$$= \int_{x_{2:} \in [0,1]^{n-1}} 1 - \cos^2(2\pi \langle x_{2:}, w_{2:} \rangle)$$

$$= \frac{1}{2}$$

Thus, $(*) = (**)$ and the integral is 0 as desired. If the condition that $w_i = z_i$ for all $i \geq 2$ does not hold, then without loss of generality $w_2 = -z_i$. By induction, $(*) = 0$, since the assumption that $w \neq z$ implies that eventually, there will be some cutoff point in indices beyond which $w$ and $z$ are exactly equal. Similarly, by applying the same trigonometric identities and $\sin(a+b) = \sin a \cos b + \cos a \sin b$ and using the induction hypothesis, $(**) = 0$ as well. Therefore, the complete sum is equal to 0, and the functions are orthogonal as desired. ■

Observe that since sin is merely a $\frac{\pi}{2}$ shift of cos, Lemma 14 also implies that the sin basis functions are orthogonal.

Recall that we decomposed the symmetric periodic function $f$ as

$$f(\boldsymbol{w}) = \sum_{[\boldsymbol{w}] \in \mathcal{O}} \sum_{\boldsymbol{w} \in [\boldsymbol{w}]} \mathrm{Re}(\hat{f}(\boldsymbol{w})) \cos(2\pi \langle \boldsymbol{x}, \boldsymbol{w} \rangle) - \mathrm{Im}(\hat{f}(\boldsymbol{w})) \sin(2\pi \langle \boldsymbol{x}, \boldsymbol{w} \rangle).$$

As in the original Barron's Theorem, we can sample frequencies of $f$. However, sampling from $w \in \mathbb{Z}^n$ will yield a function that is not necessarily invariant. Instead, if we sample *orbits* of $w$, we will obtain an invariant approximation to $f$. Concretely, for the even part of $f$, sample functions of the form $v_{[\boldsymbol{w}]} := \sum_{\boldsymbol{w} \in [\boldsymbol{w}]} sgn(\hat{f}(\boldsymbol{w})) \cos(2\pi \langle x, w \rangle)$, which are indexed by $[w] \in \mathcal{O}$, proportionally to $|\Re(\hat{f}([w]))|$. For the odd part of $f$, separately sample functions of the form $\sum_{\boldsymbol{w} \in [\boldsymbol{w}]} sgn(\hat{f}(\boldsymbol{w})) \sin(2\pi \langle x, w \rangle)$, which are indexed by $[w] \in \mathcal{O}$, proportionally to $|\mathrm{Imag}(\hat{f}([w]))|$. Note that the individual approximations are invariant, so their sum is as well, and the resultant approximation error is at most the sum of the individual approximation errors. For any $w_1, w_2 \in \mathbb{Z}^n$, by Lemma 14,

$$\int_{x \in [0,1]^n} \cos(2\pi \langle x, w_1 \rangle) \cos(2\pi \langle x, w_2 \rangle) = 0 \tag{2}$$

so long as $w_1 \neq w_2$ and $w_1 \neq w_2$. Similarly,

$$\int_{x \in [0,1]^n} \sin(2\pi \langle x, w_1 \rangle) \sin(2\pi \langle x, w_2 \rangle) = 0 \tag{3}$$

under the same conditions.

We may thus approximate $\hat{f}$ by sampling orbits over $G$ in the frequency domain. By 2, these orbits consist of sums of orthogonal functions, unless $\boldsymbol{w}$ and $-\boldsymbol{w}$ happen to be in the

same orbit in the support of $\hat{f}$. This cannot happen for the odd part of $f$, as $\hat{f}(\boldsymbol{w}) = \hat{f}(-\boldsymbol{w})$ by invariance (in this case) and $\hat{f}(\boldsymbol{w}) = \overline{\hat{f}(-\boldsymbol{w})}$ by the realness of $f$ together imply that $\hat{f}(\boldsymbol{w})$ is real. For the even part of $f$, the basis functions for $\boldsymbol{w}$ and $-\boldsymbol{w}$ are precisely the same, and so the squared norm of the orbit is still easy to analyze (and still equal to the size of the orbit). Here, the inner product between two real-valued functions is defined by

$$\Big\langle f_1(x), f_2(x) \Big\rangle = \int_{x \in [0,1]^n} f_1(x) f_2(x)$$

As a result of the orthogonality properties above and the Pythagorean theorem, we have

$$||v_{[\boldsymbol{w}]}||_2^2 = |[\boldsymbol{w}]| \tag{4}$$

Consider now the upper bound provided by Maurey's Lemma:

**Lemma 15** *Let $x$ lie in a convex set $\mathcal{V}$. Then there exist $k$ elements $v_1, \ldots, v_k \in \mathcal{V}$ and weights $\alpha_1, \ldots, \alpha_k$ such that*

$$||x - \sum_{i=1}^{k} \alpha_i v_i||^2 \leq \frac{\mathbb{E}[||x_1||^2]}{k},$$

*where $x$ is drawn from $\mathcal{V}$ proportionally to the convex coefficients of $x$.*

**Proof** Since $x \in \mathcal{V}$, there exists a random variable $u$ taking values in $\mathcal{V}$ such that $\mathbb{E}[u] = x$. Consider the squared error from approximating $\mathbb{E}[u] = x$ with $k$ random draws of variables $x_1, \ldots, x_k \sim u$:

$$\begin{aligned} \mathbb{E}[||x - \frac{1}{k} \sum_{i=1}^{k} x_i||^2] &= \frac{\sum_{i=1}^{k} \mathbb{E}[||x_i - x||^2]}{k^2} \\ &= \frac{\mathbb{E}[||x_1 - x||^2]}{k} \\ &= \frac{\mathbb{E}[||x_1||^2] + ||x||^2 - 2x^T \mathbb{E}[x_1]}{k} \\ &= \frac{\mathbb{E}[||x_1||^2] - ||x||^2}{k} \\ &\leq \frac{\mathbb{E}[||x_1||^2]}{k} \end{aligned}$$

Therefore, there exists some drawing of the variables $x_1, \ldots, x_k$ satisfying this bound; the lemma follows. ∎

Applied as in the original Barron's Theorem, Maurey's Lemma would construct $\mathcal{V}$ as $\{\mathcal{N} \cdot v_w\}$, where $v_w$ equals either $\cos(2\pi \langle x, w \rangle)$ (to approximate the even part of $f$) or $\sin(2\pi \langle x, w \rangle)$ (to approximate the odd part of $f$) up to the appropriate sign flip (according to $\hat{f}$), $p_{\boldsymbol{w}}$ is defined to be either $|\Re(\hat{f}(\boldsymbol{w}))|$ (to approximate the even part of $f$) or $|\text{Imag}(\hat{f}(\boldsymbol{w}))|$ (for the odd part), and $\mathcal{N}$ is a scaling factor, $\mathcal{N} = \sum_{\boldsymbol{w} \in \mathbb{Z}^n} p_{\boldsymbol{w}} = \sum_{[\boldsymbol{w}] \in \mathcal{C}} p_{\boldsymbol{w}} \cdot |[\boldsymbol{w}]|.$[6] Then,

---

6. Note that $v_w$ is an individual frequency, while $v_{[w]}$ denotes a summed orbit of frequencies.

$f = \sum_{\boldsymbol{w} \in \mathbb{Z}^n} p_{\boldsymbol{w}} v_w = \sum_{\boldsymbol{w} \in \mathbb{Z}^n} \frac{p_{\boldsymbol{w}}}{\mathcal{N}} (\mathcal{N} v_w)$ is a convex combination of elements of $\mathcal{V}$. The numerator of the squared $\ell_2$ error bound given by Maurey's Lemma is then:

$$
\begin{aligned}
\mathbb{E}[||x||^2] &= \sum_{\boldsymbol{w} \in \mathbb{Z}^n} \frac{p_{\boldsymbol{w}}}{\mathcal{N}} ||\mathcal{N} v_w||_2^2 \\
&= \sum_{\boldsymbol{w} \in \mathbb{Z}^n} \mathcal{N} p_{\boldsymbol{w}} \cdot ||v_w||_2^2 \\
&= \sum_{[\boldsymbol{w}] \in \mathcal{O}} \mathcal{N} p_{[\boldsymbol{w}]} \sum_{\boldsymbol{w} \in [\boldsymbol{w}]} ||v_{\boldsymbol{w}}||_2^2 \\
&= \sum_{[\boldsymbol{w}] \in \mathcal{O}} \mathcal{N} p_{[\boldsymbol{w}]} \cdot |[\boldsymbol{w}]| \qquad (5) \\
&= \mathcal{N}^2
\end{aligned}
$$

We now apply Maurey's Lemma with $\mathcal{V} = \{\mathcal{N} \cdot v_{[\boldsymbol{w}]} : [\boldsymbol{w}] \in \mathcal{O}\}$, where

$$
f = \sum_{\boldsymbol{w} \in \mathbb{Z}^n} \frac{|\hat{f}(\boldsymbol{w})|}{\mathcal{N}} (\mathcal{N} v_w) = \sum_{[\boldsymbol{w}] \in \mathcal{O}} \frac{p_{[\boldsymbol{w}]} \cdot |[\boldsymbol{w}]|}{\mathcal{N}} \Big( \sum_{\boldsymbol{w} \in [\boldsymbol{w}]} \frac{\mathcal{N}}{|[\boldsymbol{w}]|} v_{\boldsymbol{w}} \Big).
$$

Note that $\frac{p_{[\boldsymbol{w}]} \cdot |[\boldsymbol{w}]|}{\mathcal{N}}$ is a measure: $\sum_{[\boldsymbol{w}] \in \mathcal{O}} \frac{p_{[\boldsymbol{w}]} \cdot |[\boldsymbol{w}]|}{\mathcal{N}} = \sum_{[\boldsymbol{w}] \in \mathcal{O}} \sum_{\boldsymbol{w} \in [\boldsymbol{w}]} \frac{p_{[\boldsymbol{w}]}}{\mathcal{N}} = 1$. The term $\mathbb{E}[||x||_2^2]$ is given by:

$$
\begin{aligned}
\mathbb{E}[||x||^2] &= \sum_{[\boldsymbol{w}] \in \mathcal{O}} \frac{p_{[\boldsymbol{w}]} \cdot |[\boldsymbol{w}]|}{\mathcal{N}} \Big|\Big| \sum_{\boldsymbol{w} \in [\boldsymbol{w}]} \frac{\mathcal{N}}{|[\boldsymbol{w}]|} v_{\boldsymbol{w}} \Big|\Big|_2^2 \\
&= \sum_{[\boldsymbol{w}] \in \mathcal{O}} \frac{\mathcal{N} p_{[\boldsymbol{w}]}}{|[\boldsymbol{w}]|} \Big|\Big| \sum_{\boldsymbol{w} \in [\boldsymbol{w}]} v_{\boldsymbol{w}} \Big|\Big|_2^2 \\
&= \sum_{[\boldsymbol{w}] \in \mathcal{O}} \frac{\mathcal{N} p_{[\boldsymbol{w}]}}{|[\boldsymbol{w}]|} \sum_{\boldsymbol{w} \in [\boldsymbol{w}]} ||v_{\boldsymbol{w}}||_2^2 \\
&= \sum_{[\boldsymbol{w}] \in \mathcal{O}} \frac{\mathcal{N} p_{[\boldsymbol{w}]}}{|[\boldsymbol{w}]|} \cdot |[\boldsymbol{w}]| \\
&= \sum_{[\boldsymbol{w}] \in \mathcal{O}} \mathcal{N} |\hat{f}([\boldsymbol{w}])| \qquad (6)
\end{aligned}
$$

Note that $\mathcal{N}$ as defined in this section, is upper-bounded by the way it is defined in the main body (as $\sum_{\boldsymbol{w}} |\hat{f}(\boldsymbol{w})|$), so the upper bound stated in Theorem 2 holds. Comparing equations 6 and 5, it is clear that the improvement in the number of samples depends on the group $G$, and the support of $f$ in Fourier space. For example, if $\hat{f}$ is supported only on orbits of equal size $s$, where $1 \le s \le |G|$, then 5 — what we'll call "original" Barron's theorem — is a factor of $s$ larger than 6. In other words, the error induced by sampling $k$ functions ($v_{\boldsymbol{w}}$ or $v_{[\boldsymbol{w}]}$) is a factor of $s$ smaller when sampling $v_{[\boldsymbol{w}]}$, rather than $v_{\boldsymbol{w}}$. However, note that as neural network modules, $v_{[\boldsymbol{w}]}$ is always a factor of $|[\boldsymbol{w}]| \le |G|$ larger than $v_{\boldsymbol{w}}$: in particular, $v_{[\boldsymbol{w}]} = \sum_{\boldsymbol{w} \in [\boldsymbol{w}]} \frac{\mathcal{N}}{|[\boldsymbol{w}]|} v_{\boldsymbol{w}}$. (For particular groups, it may be possible to implement

this summation over an orbit more efficiently, but a naive implementation multiplies the network size by $|[\boldsymbol{w}]|$.) Therefore, a function (supported only on orbits of size $s$ in Fourier space) that can be approximated to error $\epsilon$ using $k$ individual frequencies, as in Barron's Theorem, can be approximated with an *invariant* neural network with $sk$ nodes (or $k$ orbit-averaged nodes) to error $\frac{\epsilon}{s}$. Note that for $k > s$, this implies that Barron's Theorem can be proven for *invariant* architectures of equal size (at least, when the function is supported on equal-sized orbits in Fourier space). In contrast, consider naively averaging the output of Barron's Theorem over the group $G$ produces an invariant architecture, but at a factor of $|G|$ increase in network size.

## A.4. Proof of Theorem 5

When the function is supported on unequally sized orbits in Fourier space, the previous analysis does not provide satisfactory control of the size of the network. This is because the probabilistic method used in Maurey's Lemma only proves the existence of $k$ functions $v_{[\boldsymbol{w}]}$, but their sizes may vary depending on the orbit size $|[\boldsymbol{w}]|$. Instead, as a more precise analysis, we can evaluate the following quantity, which is the product of the approximation error and the network size. Here, let $s_i$ denote the size of the orbit selected by the random variable $x_i$. Recall that we now assume $f$ is only even, such that $\hat{f}$ is real-valued, for convenience; however, one can apply the same analysis to the odd part of $f$ separately, and combine the bounds.

$$
\begin{aligned}
\mathbb{E}\left[\left\|x - \frac{1}{k}\sum_{i=1}^{k} x_i\right\|^2 \left(\sum_{i=1}^{k} s_i\right)\right] &= \frac{1}{k^2}\mathbb{E}\left[\left(\sum_{i,j=1}^{k} \langle x - x_i, x - x_j\rangle\right)\left(\sum_{\ell=1}^{k} s_\ell\right)\right] \\
&= \frac{1}{k^2}\sum_{i,j=1}^{k}\mathbb{E}\left[\langle x - x_i, x - x_j\rangle\left(\sum_{\ell=1}^{k} s_\ell\right)\right] \\
&= \frac{1}{k^2}\sum_{i,j=1}^{k}\mathbb{E}\left[\langle x - x_i, x - x_j\rangle\left(s_i + s_j + \sum_{\ell\neq i,j} s_\ell\right)\right] \\
&= \frac{1}{k^2}\sum_{i,j=1}^{k}\mathbb{E}\left[\langle x - x_i, x - x_j\rangle\left(s_i + s_j + \sum_{\ell\neq i,j} s_\ell\right)\right] \\
&= \frac{1}{k^2}\sum_{i,j=1}^{k}\mathbb{E}\left[\langle x - x_i, x - x_j\rangle\left(s_i + s_j\right)\right] \\
&\quad + \mathbb{E}\left[\left(\sum_{\ell\neq i,j} s_\ell\right)\right]\mathbb{E}\left[\langle x - x_i, x - x_j\rangle\right]
\end{aligned}
\tag{7}
$$

Note that $\mathbb{E}[\langle x - x_i, x - x_j\rangle] = \|x\|^2 - 2\|x\|^2 + \|x\|^2 = 0$ for $i \neq j$. Similarly, when $i \neq j$:

$$
\begin{aligned}
\mathbb{E}[s_i\langle x - x_i, x - x_j\rangle] &= \mathbb{E}[s_i]\|x\|^2 - \mathbb{E}[s_i]x^T\mathbb{E}[x_j] - x^T\mathbb{E}[s_ix_i] + \mathbb{E}[x_j^T]\mathbb{E}[s_ix_i] \\
&= \mathbb{E}[s_i]\|x\|^2 - \mathbb{E}[s_i]\|x\|^2 - x^T\mathbb{E}[s_ix_i] + x^T\mathbb{E}[s_ix_i] \qquad = 0
\end{aligned}
$$

Therefore, (7) vanishes when $i \neq j$, and defining the constant value $S_{k-1} := \mathbb{E}[\sum_{\ell \neq i} s_\ell]$, we can simplify (7) to:

$$\frac{1}{k^2} \sum_{i,j=1}^{k} \mathbb{E}\left[\langle x - x_i, x - x_j \rangle \left(s_i + s_j\right)\right] + \mathbb{E}\left[\left(\sum_{\ell \neq i,j} s_\ell\right)\right] \mathbb{E}\left[\langle x - x_i, x - x_j \rangle\right]$$

$$= \frac{1}{k^2} \sum_{i=1}^{k} \mathbb{E}[||x - x_i||^2 s_i] + S_{k-1}\mathbb{E}[||x - x_i||^2]$$

$$= \frac{1}{k^2} \sum_{i=1}^{k} \mathbb{E}[s_i]||x||^2 - 2\langle x, \mathbb{E}[x_i s_i]\rangle + \mathbb{E}[s_i ||x_i||^2] + S_{k-1}||x||^2 - 2S_{k-1}||x||^2 + S_{k-1}\mathbb{E}[||x_i||^2]$$

$$= \frac{1}{k^2} \sum_{i=1}^{k} \mathbb{E}[s_i]||x||^2 - 2\langle x, \mathbb{E}[x_i s_i]\rangle + \mathbb{E}[s_i ||x_i||^2] - S_{k-1}||x||^2 + S_{k-1}\mathbb{E}[||x_i||^2]$$

$$= \frac{1}{k} \left[\mathbb{E}[s_1]||x||^2 - 2\langle x, \mathbb{E}[x_1 s_1]\rangle + \mathbb{E}[s_1 ||x_1||^2] - S_{k-1}||x||^2 + S_{k-1}\mathbb{E}[||x_1||^2]\right]$$

Finally, note that

$$\mathbb{E}[s_1] = \sum_{[\boldsymbol{w}]\in\mathcal{O}} \frac{|\hat{f}([\boldsymbol{w}])| \cdot |[\boldsymbol{w}]|^2}{\mathcal{N}}, \tag{8}$$

and moreover,

$$\mathbb{E}[||x_1||^2] = \sum_{[\boldsymbol{w}]\in\mathcal{O}} \frac{|\hat{f}([\boldsymbol{w}])| \cdot |[\boldsymbol{w}]|}{\mathcal{N}}||\mathcal{N} v_{[\boldsymbol{w}]}||^2$$

$$= \mathcal{N} \sum_{[\boldsymbol{w}]\in\mathcal{O}} p_{[\boldsymbol{w}]} \cdot |[\boldsymbol{w}]|||v_{[\boldsymbol{w}]}||^2$$

$$= \mathcal{N} \sum_{[\boldsymbol{w}]\in\mathcal{O}} |\hat{f}([\boldsymbol{w}])| \cdot |[\boldsymbol{w}]|^2$$

$$= \mathcal{N}^2 \mathbb{E}[s_1].$$

Similarly,

$$\mathbb{E}[s_1 ||x_1||^2] = \sum_{[\boldsymbol{w}]\in\mathcal{O}} \frac{|\hat{f}([\boldsymbol{w}])| \cdot |[\boldsymbol{w}]|^2}{\mathcal{N}}||\mathcal{N} v_{[\boldsymbol{w}]}||^2$$

$$= \mathcal{N} \sum_{[\boldsymbol{w}]\in\mathcal{O}} |\hat{f}([\boldsymbol{w}])| \cdot |[\boldsymbol{w}]|^2 ||v_{[\boldsymbol{w}]}||^2$$

$$= \mathcal{N} \sum_{[\boldsymbol{w}]\in\mathcal{O}} |\hat{f}([\boldsymbol{w}])| \cdot |[\boldsymbol{w}]|^3$$

$$= \mathcal{N}^2 \mathbb{E}[s_1^2]$$

Combining with $S_{k-1} = (k-1)\mathbb{E}[s_1]$, we finally obtain

$$\mathbb{E}[s_1]||x||^2 - 2\langle x, \mathbb{E}[x_1 s_1]\rangle + \mathbb{E}[s_1||x_1||^2] - S_{k-1}||x||^2 + S_{k-1}\mathbb{E}[||x_1||^2]$$
$$=\mathbb{E}[s_1]||x||^2 - 2\langle x, \mathbb{E}[x_1 s_1]\rangle + \mathbb{E}[s_1||x_1||^2] - (k-1)\mathbb{E}[s_1]||x||^2 + (k-1)\mathbb{E}[s_1]\mathcal{N}^2\mathbb{E}[s_1]$$
$$=(2-k)\mathbb{E}[s_1]||x||^2 - 2\langle x, \mathbb{E}[x_1 s_1]\rangle + \mathcal{N}^2\mathbb{E}[s_1^2] + (k-1)\mathbb{E}[s_1]^2\mathcal{N}^2$$

Note that, when $s_1$ is a constant, the first two terms simplify to a single negative term; in the upper bound from Maurey's lemma, this term is dropped.

## A.5. Group-averaged Barron's theorem

We begin with the case of invariance, and have the following theorem.

**Theorem 16** *Let $G$ be a finite subgroup of $O(n)$ and let $\mathcal{X} \subseteq \mathbb{R}^n$ be a space[7] on which $G$ acts. Furthermore, suppose $g\mathcal{X} = \mathcal{X}$ for all $g \in G$. Let $f : \mathcal{X} \to \mathbb{R}$ be a $G$-invariant function. Then there exists a two-layer group convolutional network $NN_{G-CNN}$, with sigmoidal nonlinearity, $\Theta(|G| \cdot k)$ units, and approximation error bounded as follows:*

$$\int_{x \in \mathcal{X}} |f(x) - NN_{G-CNN}(x)|^2 \leq \frac{C_f}{k}$$

*Here, $C_f$ is the Barron constant: $C_f = \int_{w \in \mathbb{R}^n} ||w|| \cdot |\hat{f}(w)|$, minimized over all extensions of $f$ to $\mathbb{R}^n$.*

**Remark 17** *In contrast to existing invariant universal approximation results, such as those of Bogatskiy et al. (2020) and Yarotsky (2018), Theorem 16 provides an explicit and moderate bound on the size of the approximating networks in terms of the group size. It avoids both exponential dependence on dimension, which arises in any blackbox call to traditional approximation results, and any dependence on the number of invariants needed for Hilbert's theorem (both of which are central to Yarotsky (2018) and the works which build upon it.)*

We now prove Theorem 16.
**Proof**

Let $\mathcal{X}$ be a non-homogeneous space on which $G$, a subgroup of the orthogonal group, acts, and suppose $g\mathcal{X} = \mathcal{X}$ for all $g \in G$. Let $f : \mathcal{X} \to \mathbb{R}$. Moreover, suppose $f$ is invariant, i.e. $f(gx) = f(x)$ for all $g \in G$, $x \in \mathcal{X}$. By Barron's Theorem, for any $\epsilon$ there exists a depth two ReLU net

$$NN_{Barron}(x) = \boldsymbol{\alpha}^T \sigma(Wx - b)$$

with $h = O(\frac{C_f}{\epsilon})$ hidden units (i.e. $\boldsymbol{\alpha} \in \mathbb{R}^h$) such that

$$\int |f(x) - NN_{Barron}(x)|^2 dx \leq \epsilon$$

---

7. If $\mathcal{X}$ is a homogeneous space, then $f$ is a trivial function, so we think of $\mathcal{X}$ as a non-homogeneous space on which $G$ acts.

where $C_f = \left( \int ||w|| \cdot |\hat{f}(w)| dw \right)^2$. Call the $i^{th}$ row of the matrix $W$ $w_i^T$. Then define the following group-convolutional network:

$$NN_{G-CNN}(x) = \frac{1}{|G|} \sum_i \boldsymbol{\alpha}_i \sum_{g \in G} \sigma((w_i * x)(g) - b_i)$$

$NN_{G-CNN}$ is an $h$-channel group convolutional network with a sigmoidal nonlinearity, which pools over both group elements and channels.

$$
\begin{aligned}
NN_{G-CNN}(x) &= \frac{1}{|G|} \sum_i \boldsymbol{\alpha}_i \sum_{g \in G} \sigma((w_i * x)(g) - b_i) \\
&= \frac{1}{|G|} \sum_{g \in G} \sum_i \boldsymbol{\alpha}_i \sigma(w_i^T g x - b_i) \\
&= \frac{1}{|G|} \sum_{g \in G} \boldsymbol{\alpha}^T \sigma(W g x - b) \\
&= \frac{1}{|G|} \sum_{g \in G} NN_{Barron}(gx)
\end{aligned}
$$

For any function $f$, let $f^g(x) = f(gx)$. Although $NN_{Barron}(gx) \neq NN_{Barron}(x)$, we can extend its error guarantee to $NN_{G-CNN}$ as follows:

$$
\begin{aligned}
\left( \int |f(x) - NN_{G-CNN}(x)|^2 \right)^{1/2} dx &= ||f - NN_{G-CNN}|| \\
&= ||\frac{1}{|G|} \sum_{g \in G} f^g - \frac{1}{|G|} \sum_{g \in G} NN_{Barron}^g|| \\
&\leq \frac{1}{|G|} \sum_{g \in G} ||f^g - NN_{Barron}^g|| \\
&= \frac{1}{|G|} \sum_{g \in G} ||f - NN_{Barron}|| \\
&\leq \sqrt{\epsilon}
\end{aligned}
$$

Here, we used Jensen's inequality and the integration change of variable $x \mapsto gx$, enabled by the orthogonality of $g$ and the assumption $\mathcal{X} = g\mathcal{X}$ for all $g$. ∎

By appropriately modifying the final linear layer of the $NN_{G-CNN}$ architecture described above, we obtain the following analogous theorem for *equivariant* functions.

**Theorem 18**  *Let $G$ be a finite subgroup of $O(n)$ acting orthogonally on $\mathbb{R}^n$, as well as acting on $\mathbb{R}^m$ via a linear representation $\rho$. Let $f : \mathbb{R}^n \to \mathbb{R}^m$ be a $G$-equivariant function*[8].

---

8. As in Theorem 16, we could again state this theorem for subsets of $\mathbb{R}^n$ and $\mathbb{R}^m$ on which $G$ acts, but do not do so for simplicity.

Then there exists a two-layer group convolutional network $NN_{G-CNN}$, with sigmoidal non-linearity, with $\Theta(|G| \cdot k)$ units and approximation error bounded as follows:

$$\int_{x \in \mathbb{R}^n} \|f(x) - NN_{G-CNN}(x)\|^2 \leq \frac{D^2}{k} \sum_s C_f^s$$

Here, $D$ is a uniform bound on the operator norm of the matrix $\rho(g)$ over all $g \in G$, and $C_f^s$ is the Barron parameter of $f_s$.

**Proof** The architecture of Theorem 16 averages the approximation given by Barron's Theorem over the group. For the equivariant case, it is then natural to approximate each output dimension by Barron's Theorem, and then to average in an equivariant way. This will contribute an additional layer, to apply the representation $\rho$.

Formally, begin by approximating the $s^{th}$ output of $f$ as $NN_{Barron}^s$, defined by

$$NN_{Barron}(x)^s = (\alpha^s)^T \sigma(W^s x - b^s) = \sum_i \alpha_i^s \sigma(\langle w_i^s, x \rangle - b^s).$$

Letting $C_f^s$ denote the Barron parameter of $f_s$, $NN_{Barron}^s$ has $\Theta(k)$ hidden nodes and squared $L_2$ approximation error upper bounded by $\Theta(\frac{C_f^s}{k})$. Then, define $NN_{G-CNN}$ as follows:

$$NN_{G-CNN}(x) := \frac{1}{|G|} \sum_{g \in G} \rho(g^{-1}) NN_{Barron}(gx)$$

One can verify that $NN_{G-CNN}$ is equivariant:

$$\begin{aligned}
NN_{G-CNN}(hx) &= \frac{1}{|G|} \sum_{g \in G} \rho(g^{-1}) NN_{Barron}(ghx) \\
&= \frac{1}{|G|} \sum_{k \in G} \rho(hk^{-1}) NN_{Barron}(kx) \\
&= \frac{1}{|G|} \sum_{k \in G} \rho(h) \rho(k^{-1}) NN_{Barron}(kx) \\
&= \rho(h) NN_{G-CNN}(x)
\end{aligned}$$

Moreover, its error is bounded as follows:

$$
\left( \int_{x \in \mathbb{R}^n} \| f(x) - NN_{G-CNN}(x) \|^2 \right)^{1/2} = \left( \int_{x \in \mathbb{R}^n} \left\| f(x) - \frac{1}{|G|} \sum_{g \in G} \rho(g^{-1}) NN_{Barron}(gx) \right\|^2 \right)^{1/2}
$$

$$
= \left( \int_{x \in \mathbb{R}^n} \left\| \frac{1}{|G|} \sum_{g \in G} \left( \rho(g^{-1}) f(gx) - \rho(g^{-1}) NN_{Barron}(gx) \right) \right\|^2 \right)^{1/2}
$$

$$
\leq \frac{1}{|G|} \sum_{g \in G} \left( \int_{x \in \mathbb{R}^n} \left\| \rho(g^{-1}) f(gx) - \rho(g^{-1}) NN_{Barron}(gx) \right\|^2 \right)^{1/2}
$$

$$
\leq \frac{D}{|G|} \sum_{g \in G} \left( \int_{x \in \mathbb{R}^n} \left\| f(gx) - NN_{Barron}(gx) \right\|^2 \right)^{1/2}
$$

$$
= \frac{D}{|G|} \sum_{g \in G} \left( \int_{x \in \mathbb{R}^n} \left\| f(x) - NN_{Barron}(x) \right\|^2 \right)^{1/2}
$$

$$
\leq D \left( \sum_{s=1}^{m} \int_{x \in \mathbb{R}^n} \left| f^s(x) - NN_{Barron}^s(x) \right|^2 \right)^{1/2}
$$

$$
\leq D \left( \sum_{s=1}^{m} \frac{C_f^s}{k} \right)^{1/2}
$$

∎

**Remark 19** *Theorem 16 and Theorem 18 are stated for finite groups, but the proofs clearly hold for infinite $G$, with each summation replaced by an integral over $G$ with respect to the Haar measure.*

As a result of Theorem 16, the approximation error of $NN_{G-CNN}$ is equal to that of $NN_{Barron}$, but it is representable as a group-convolutional network with $h$ channels. This demonstrates not only that Barron's Theorem extends to show universality of group-convolutional networks for invariant functions, but more generally that the orbit-averaged version of any two-layer net is representable as a group-convolutional net.

Unfortunately, the architecture requires convolving over $G$, which either contributes a factor of $|G|$ additional neurons for a large finite group, or is not (obviously) computable and requires special treatment, such as approximating the convolution in group Fourier space for infinite groups. However, another way of interpreting this architecture is as replacing its first nonlinear layer, $\sigma(Wx - b)$, with a group-averaged version $\frac{1}{|G|} \sum_{g \in G} \sigma(Wgx - b)$. Theorem 7 (proven in Appendix A.6) demonstrates a special case in which this expression can be computed analytically, thereby reducing the effective number of neurons in Theorem 16 from $|G| \cdot k$ to $k$.

## A.6. Proof of Theorem 7

The previous section demonstrated that any invariant function $f$ with Barron parameter $C_f$ is representable by an *invariant* neural net, whose size scales with $|G|$ but is not exponential in the input dimension. The group-averaged architecture can be understood as an ordinary two layer architecture with final pooling layer, but which crucially replaces the first layer — usually composed of a linear layer with a pointwise nonlinearity — with a group-averaged nonlinear function $\bar{\sigma}_{w,b}(x) = \int_{g \in G} \sigma(Wgx - b)$. It is this dependence on $|G|$ which contributes adversarially to the network size; if $G$ is large, such as $S_n$ with $|S_n| = n!$, or even infinite, such as $G = SO(n)$, this integral is intractable. It would therefore be of both theoretical and practical interest to devise schemes for computing $\bar{\sigma}_{w,b}(x)$ for common groups $G$ in an automatically differentiable and efficient manner.

**Failure of Polynomials** To alleviate this issue, one might hope to approximate the nonlinearity (cosine, $\sigma$, indicator, or general sigmoidal) with a polynomial over the range of inputs. This is because bounded-degree polynomials can be easily and analytically averaged over compact groups as follows. First, write a polynomial of degree $d$ in terms of its homogeneous parts (using $\cdot$ to denote inner products between higher-dimensional tensors):

$$p(x) = \sum_{i=0}^{d} C_i \cdot x^{\otimes i}.$$

Then, integration over the group takes a simple form:

$$\int_g p(gx) = \int_g \sum_{i=0}^{d} C_i \cdot (gx)^{\otimes i}$$

$$= \sum_{i=0}^{d} C_i \cdot \left( \int_g g^{\otimes i} \right) \cdot x^{\otimes i}$$

Note then that $\int_g g^{\otimes i}$ is simply the Reynolds operator corresponding to the $i^{th}$ tensor product representation of the group. Using the Clebsch-Gordan decomposition, it is possible to decompose $g^{\otimes i}$ into irreducible representations, which are easy to integrate analytically (in particular, as an orthonormal basis, they always integrate to 0 with the exception of the trivial irreducible representation). Unfortunately, it is crucial for universal approximation results to use *non-polynomial* non-linearities. By approximating a non-polynomial nonlinearity with a finite-degree polynomial, we impede universal approximation and, in fact, can approximate multidimensional invariant functions no better than a multivariate group-averaged Taylor expansion.

We return, then, to the problem of exactly computing $\bar{\sigma}_{w,b}(x)$. This will depend greatly on $\sigma$ and $G$, and in fact, may not always be possible. However, we show that for $\sigma(x) = \mathbb{1}(x \geq 0)$ and $G = SO(n)$, the function $\bar{\sigma}_{W,b}$ is easily expressible in closed form.

**Theorem 20** *When averaged over the special orthogonal group in $n$ dimensions, the non-linear invariant function $\bar{\mathbb{1}}_{w,b}(x)$ is expressible as follows:*

$$\bar{\mathbb{1}}_{w,b}(x) := \int_{g \in SO(n)} \mathbb{1}(w^T gx - b \geq 0) = \frac{1}{2} I_{\sin(\phi)^2}\left(\frac{n-1}{2}, \frac{1}{2}\right).$$

Here, $\phi = \cos^{-1}(\frac{b}{||w||\cdot||x||})$ and $I_a(b,c)$ denotes the regularized incomplete beta function.

**Proof** Consider a fixed $x$, $w$, and $b$. Define an equivalence relation on $SO(n)$ with respect to $x$: $g \sim h$ if $gx = hx$. Therefore, if $g \sim h$, then $\mathbb{1}(w^T gx - b) = \mathbb{1}(w^T hx - b)$. The equivalence classes, denoted $[g]$, of $SO(n)$, can then be associated with vectors of length equal to $||x||$. Moreover, it is possible to integrate over a given equivalence class $[g]$ using the Haar measure on $SO(n-1)$. (Intuitively, any rotation in $[g]$ simply must rotate $x$ to a specified $n$-dimensional vector, and then can perform any $n-1$ dimensional rotation in the orthogonal subspace to this vector.) Denote this set by $B(0, ||x||)$. Then, the following equality holds:

$$\int_{g \in SO(n)} \mathbb{1}(w^T gx - b) = \int_{v \in B(0,||x||)} \int_{g \in SO(n) \text{ s.t. } gx=v} \mathbb{1}(w^T gx - b \geq 0)$$

$$= \int_{v \in B(0,||x||)} \mathbb{1}(w^T v - b \geq 0)$$

Here, the integration over $B(0, ||x||)$ is with respect to the uniform probability measure on the sphere $B(0, ||x||)$ in $n$ dimensions. Moreover, the final expression has a simple geometrical interpretation, which will give rise to the stated analytic form. In particular, note the following:

$$\mathbb{1}(w^T v - b) = 1 \iff w^T v \geq b$$

$$\iff \cos \angle(w, v) \geq \frac{b}{||w|| \cdot ||v||}$$

Therefore, the desired integral is simply equal to the fraction of the surface area of $B(0, ||x||)$ lying in the spherical cap of angle $\phi := \cos^{-1}(\frac{b}{||w||\cdot||x||})$. Using the analytic form of the surface area of high-dimensional spherical caps, expressed neatly in terms of the incomplete beta function by Li (2011), yields the following expression for this integral:

$$\frac{\pi^{n/2} r^{n-1}}{\Gamma(\frac{n}{2})} I_{\sin(\phi)^2}\left(\frac{n-1}{2}, \frac{1}{2}\right)$$

Finally, recall that the surface area of an $r$-radius sphere in $n$ dimensions is given by $\frac{2\pi^{n/2} r^{n-1}}{\Gamma(n/2)}$. As a result, the fraction of $B(0, ||x||)$ lying in the relevant cap is given by

$$\frac{1}{2} I_{\sin(\phi)^2}\left(\frac{n-1}{2}, \frac{1}{2}\right)$$

$\blacksquare$

Unfortunately, the regularized incomplete beta functions are not themselves easily expressible as small neural networks with standard nonlinearities. However, this quantity is computable using predefined special functions in many programming languages. If computed in an automatically differentiable manner, this makes practical the equivariant architecture of the previous section for $SO(n)$, an infinite group — albeit with a specialized nonlinearity.

