# OpenReview forum: "Barron's Theorem for Equivariant Networks"
_NeurIPS.cc/2022/Workshop/NeurReps — NeurReps 2022 Poster_

### Official Review · Reviewer_q7Nx · 2022-10-07
**Review of Barron's Theorem for Equivariant Networks**

**Confidence:** 4
**Soundness:** 3
**Presentation:** 3
**Contribution:** 3
**Overall Rating:** 7

**Summary:**

The paper studies the design of invariant networks for the groups $\mathcal{S}_n$ and $SO(2)$. The main theory in this paper requires the use of Barrons theorem to bring down the effective parameter count of these networks such that they are comparable to their non-invariant counterparts. For the rotation group the authors also propose a novel non-linearity to aid in their construction.

**Questions:**

One question that I had is that current invariant architectures have less number of "effective" parameters (the ones being learned) than their non-invariant counterparts but in practice they still need to realize the entire parameter cost---i.e. a Group-Convolution can be implemented as a regular convolution with a special kernel. Is there a difference between the "effective" parameter cost vs. actual cost in this work?

**Limitations:**

I would like to see the following experiments in an updated paper:

1.) Experiments comparing higher order graph networks and this approach on standard graph benchmarks (e.g. Maron et. al 2019)
2.) Experiments comparing SO(2) invariant networks to G-CNNs for Rotation MNIST, Fliprot MNIST, CIFAR-10.
3.) The computation cost of using this method in terms of time complexity (e.g. time/epoch) vs. current invariant architectures.

**Recommended Decision:**

3: Accept

**Relevance:**

4: Highly relevant

**Strengths And Weaknesses:**

I think this paper is relatively well written and tackles an important problem in the invariant architecture literature. The paper is of a theoretical nature and effectively borrows a lot of tools for their intended purpose. This is by no means a bad thing as improving the parameter efficiency by a factor of $|G|$ is welcome. I should mention however, despite the impressive looking theory I did not check for correctness in the Appendix.

The paper has a few weaknesses that authors could help address in updated manuscripts. Usually in invariant machine learning there are three pillars of consideration, the data, the invariant architecture, and the optimization problem. Indeed, while these networks appear to be better in terms of number of parameters it is unclear at this point whether they are equally performant. This needs to be empirically verified and with sufficient care. For example, even a 1 layer infinite layer NN is a universal approximation but that doesn't mean we use these in practice. As a result, this paper would benefit from experiments that complement its impressive theory.



**Submission Track:**

Extended Abstract (4 Page)

---

> ### Author Response · Authors · 2022-11-02
> **Thank you for the review; a brief reply**
>
> Many thanks to the reviewer for their thoughtful comments. The reviewer is correct in pointing out the shortcomings of universal approximation results. We do not currently have any method for computing these efficient networks from data; the result is theoretical in nature, and aimed to validate the existence of networks that are at least "practically" sized for many functions. The distinction between "effective" parameters and the entire parameter cost is also an important one, and we have added a sentence on this to the camera-ready. Group-averaging any function maintains the number of "effective" or learnable parameters, but is computationally infeasible for many groups (such as the permutation group). Therefore, we consider the number of hidden nodes instead, which correspond more closely to the complexity of a forward pass. Finally, although the results on permutation subgroups are (as noted) theoretical in nature and therefore not suited to experiments, we will consider experiments on SO(n)-invariant networks in any future work in this direction.

---

### Official Review · Reviewer_AJyA · 2022-10-12
**Several references are missing which makes unclear the precise contribution of this work**

**Confidence:** 5
**Soundness:** 4
**Presentation:** 3
**Contribution:** 2
**Overall Rating:** 5

**Summary:**

This work proposes a universal approximation theorem with extra-assumption which allows to improve w.r.t. Barron's theorem.

**Questions:**

I believe it's necessary to compare with the prior works I mentionned. In particuliar, I'm unclear if this work is quantitative version of [1] or [2]; I would like to get a clarification.

**Limitations:**

The state of the art is clearly not well discussed.

**Recommended Decision:**

1: Reject

**Relevance:**

2: Limited relevance

**Strengths And Weaknesses:**

I think some comparison with state-of-the-art approaches are currently missing, e.g. [1] https://arxiv.org/pdf/1905.04943.pdf or e.g., [2] "Breaking the curse of dimensionality with convex neural networks", F. Bach, as they exploit identical ideas. I'm not sure of the relevance of Thm 7 given that the found function seems non-trivial to compute ; Thm 5 is just a standard sampling argument and the Thm 3 seems extremely similar to the setting of harmonic polynomials described in [2]. I am not sure to understand the difference and that would need to be discussed. Thus, I have a borderline opinion on this paper, leaning toward rejection.

**Submission Track:**

Extended Abstract (4 Page)

---

> ### Author Response · Authors · 2022-11-02
> **Thank you for the review; a brief reply**
>
> We thank the reviewer for their thoughtful notes, and attempt to reply briefly here. In short, to the best of our understanding, this work is not a quantitative version of [1] or [2]. Regarding [1], we have added to the camera-ready a remark on the comparison to Maron et al 2019, "On the Universality of Invariant Networks", since [1] appears to provide an alternative proof of their result (and the results of [1] on equivariance are not relevant, since we consider only invariant functions). Regarding [2], while the motivation and results seem similar to the original Barron's Theorem, it does not seem to consider invariant approximations of invariant functions, and is therefore not comparable to Theorem 2. Indeed, our primary goal was to obtain approximations which are themselves invariant, rather than to compare with all techniques for avoiding the curse of dimensionality in universal approximation more broadly. Further clarification is most certainly welcome if we have misunderstood the relevance of [1] or [2].

---

### Official Review · Reviewer_FDEa · 2022-10-16
**Barron's Theorem for Equivariant Networks**

**Confidence:** 4
**Soundness:** 3
**Presentation:** 3
**Contribution:** 3
**Overall Rating:** 7

**Summary:**

This work extends Barron's theorem on neural network function approximation to incorporate knowledge of a group acting on the domain of the function.  This allows for smaller network architectures in universality statements.  The contribution includes mathematical statements and proofs for the approach applied to permutation subgroups and SO(n).


**Questions:**

How important is the SOO condition?  What do these neural networks look like for a simple example that perhaps can be solved?  Is there a general strategy that works for compact groups more generally?  Are there any practical approaches to finding these networks from data, and have any of them been tried?  How tight is the "N" in Theorem 3 (respectively, Barron's parameters C_f)?  More specifically, are there natural function classes for which this parameter can be calculated or estimated?


**Limitations:**

Although technically an improvement to Barron's theorem in the case of additional group structure imposed on the approximating function f, it is unclear how to turn these ideas into something of practical benefit on real data and problems.  If more can be said in this regard, the work would be significantly strengthened, especially since the abstract mentions (and italicizes) that efficiency could be gained with the approach.


**Recommended Decision:**

3: Accept

**Relevance:**

4: Highly relevant

**Strengths And Weaknesses:**

The approach builds on the work of Barron to prove existence of group-invariant neural networks approximating a given function.  The main technical contribution is to allow fewer sampled orbits than required by simply "group averaging" the classical results.  Two rather general specific contexts are explained, including permutation subgroups as well as the matrix group SO(n).

The main contributions of this work are mathematical in nature, which is important for the field of machine learning, especially considering that symmetry is becoming a key factor in moving the field forward.  However, it is unclear how relevant they will be to practical applications as no experiments or data analysis is presented.  In addition, it is not clear how important is the main assumption of the analysis ("SOO condition").

Typically, such results of universality are technical statements and not intended to be used in practice.  They only suggest that such neural networks exist in theory, which nonetheless does inform practitioners what is possible.


**Submission Track:**

Extended Abstract (4 Page)

---

> ### Author Response · Authors · 2022-11-02
> **Thank you for the review; a brief reply**
>
> We thank the reviewer for their positive review and comments on the paper. Regarding the SOO condition, it was a minor technical condition, which we were in fact able to remove for the camera-ready version. Unfortunately, since Barron's Theorem does not admit a practical method for computing such networks from data, neither does this result for permutation subgroups; it is purely theoretical in nature. We will consider the reviewer's other helpful suggestions (considering whether N is tight, estimating it for natural function classes, and considering the practical benefit in the case of SO(n)) going forward.

---

### Decision · Program_Chairs · 2022-10-21

Accept (Poster)